# The Difficulty Detecting Tuberculosis in a Child with Post-COVID-19 and Cerebral Palsy

**DOI:** 10.3390/diagnostics13172826

**Published:** 2023-08-31

**Authors:** Andjelka Stojkovic, Irena Ilic, Andrijana Kostic, Katerina Dajic, Zorica Raskovic, Jelena Nestorovic, Milena Ilic

**Affiliations:** 1University Clinical Center Clinic of Pediatrics Kragujevac Serbia, 34000 Kragujevac, Serbia; andrijanak88@yahoo.com (A.K.); katerinadajic@gmail.com (K.D.); drzoricaraskovic@gmail.com (Z.R.); jelenanestorovickg@gmail.com (J.N.); 2University of Kragujevac, Serbia, Faculty of Medical Sciences, Department of Pediatrics, 34000 Kragujevac, Serbia; 3Department of Epidemiology, Faculty of Medicine University of Belgrade, 11000 Belgrade, Serbia; irena.ilic@med.bg.ac.rs; 4University of Kragujevac, Serbia, Faculty of Medical Sciences, Department of Epidemiology, 34000 Kragujevac, Serbia; drmilenailic@yahoo.com

**Keywords:** prolonged COVID-19, tuberculosis, cerebral palsy, examinations and diagnoses

## Abstract

When hypostatic pneumonia is present at the same time as COVID-19 pneumonia, the clinical course is almost always prolonged (prolonged-COVID-19) due to persistent inflammation, long-term anti-inflammatory syndrome, followed by immune exhaustion, i.e., by immunosuppression and catabolic syndrome. In the immunosuppression phase, viral reactivation can be accompanied by a secondary infection, which, in this case, is pulmonary tuberculosis. Pulmonary tuberculosis in post-COVID-19 patients and in patients with spastic quadriplegic cerebral palsy does not have a typical clinical course nor laboratory, radiological, immunological, microbiological, or fiberbronchoscopic pathohistological confirmation. Due to this, the treatment of pulmonary tuberculosis was not carried out on time, postponed after the unsuccessful treatment of sepsis, post-COVID-19, and other accompanying viral (adenovirus, RSV) and bacterial (streptococcus viridans) infections. The treatment of pulmonary tuberculosis was possible only “ex juvantibus” (trial) post-COVID-19. It becomes imperative to search for a new, more precise and reliable diagnostic test for the detection of tuberculosis bacillus.

A boy with a body weigh of 15 kg, who has spastic quadriplegic cerebral palsy (SQCP), was down clinically inapparent COVID-19 which we confirmed due to an elevated titer of IgG antibodies for SARS-CoV-2 (40.6 BAU/mL) on admission to the Clinic of Paediatrics (PC-Kg) (Table 1). The child was referred to PC-Kg due to experiencing intermittent fever during the previous 3 months, but with a continuous fever 2 weeks before admission to PC-Kg. During the last 2 weeks, he was treated with antibiotics (semi-synthetic penicillin, then third-generation cephalosporin, and the day before admission, azithromycin) via bronchodilator inhalation, so that C-reactive protein (CRP) fell to 114 mg/L, compared to the previous level (305 mg/L). According to the anamnestic data of the last 3 months, the patient had no weight loss, no sweating, no cough “with a full mouth”, no history of tuberculosis in the family and environment, and showed no evidence of suffering from COVID-19. The Quantiferon TB gold immunoassay (interferon-γ release assay, IGRA) result was negative, and there was no increase in total serum immunoglobulin M nor decrease in total serum immunoglobulin A during the 34 days of hospitalization, due to the patient’s suppressed immune response during the period of post-COVID-19 (Table 1) [1]. A negative tuberculin skin test (TST) is a consequence of suppression by systemic corticosteroids (Table 1). The direct microscopy of all three samples of tracheal aspirate and gastric washings, after staining with Ziehl-Neelsen, on the 28th, 29th and 30th day of hospitalization did not find tuberculosis bacilli, nor did the analysis of the fiber-aspirate sample on the 36th day of hospitalization (Institute for Mother and Child New Belgrade (IMC) (Table 1). Only 45 days after the start of “ex juvantibus” treatment with antituberculotics (H,R,Z,E), was pulmonary tuberculosis confirmed by microbiological examination of the tracheal aspirate and gastric washings in a Loweinsten medium, in a patient with post-COVID-19 and SQCP [2,3,4,5]. Other results were within the reference values: urine, blood gas analysis, erythrocyte count in blood test, prothrombin time, activated partial thromboplastin time, transaminases (AST, ALT), gamma-GT, glycemia, urea, creatinine, alkaline phosphatase, creatinine kinase (total, muscle), urine culture, virological serology for cytomegalovirus, toxoplasma, rubella, herpes simplex 1 + 2, parvo-B19, Epstein-Barr virus (Table 1). Streptococcus viridans was isolated in 1 blood culture, but 11 blood cultures with two samples were left without pathogenic germs (Table 1). Adenovirus and respiratory syncytial virus (RSV) have been proven via polymerase chain reaction (PCR) diagnostics for the 17 most common respiratory pathogens (Table 1) [6,7,8].

**Figure 1 diagnostics-13-02826-f001:**
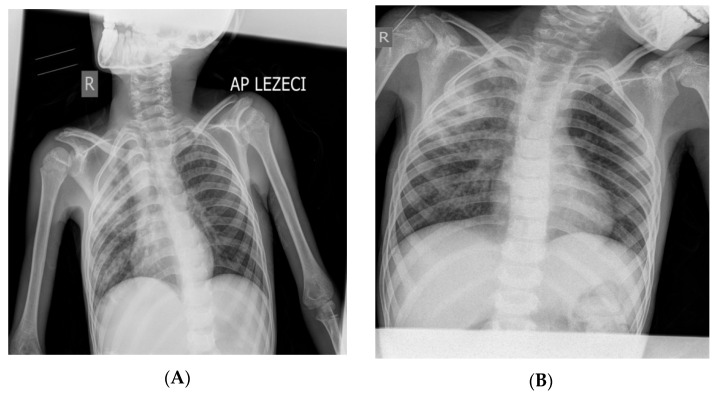
X-ray of the boy’s lungs on admission (**A**) and after 28 days of hospitalization (**B**). (**A**) X-rays of the lungs (in the supine position) were performed on admission: right (“R”) diffusely spotted shadows confluent in the upper and middle lung field, left hiloapically spotted shadows. Cardiophrenic sinuses free, both hemidiaphragm clearly contoured (Figure 1). (**B**) X-rays of the lungs (in the supine position) were performed on the 28th day of hospitalization: diffusely accentuated interstitium with a zone of consolidation in the upper and middle pulmonary field on the right (“R”). Hemidiaphragms are clearly contoured (Figure 1). The pediatric scoring system A for pulmonary tuberculosis was not sufficient to confirm pulmonary tuberculosis (mark 3) [9,10,11,12,13]. During the 34 days of hospitalization, the boy was febrile all the time with occasional temperature jumps up to 39.5 °C (septic type); he was breathing stably throughout, without the need for oxygen therapy. Only the patient’s fever, which manifested itself continuously for 4 weeks, raised the suspicion of pulmonary tuberculosis. The cough was intermittent and discreet during the 34 days of hospitalization in PC-Kg.

**Figure 2 diagnostics-13-02826-f002:**
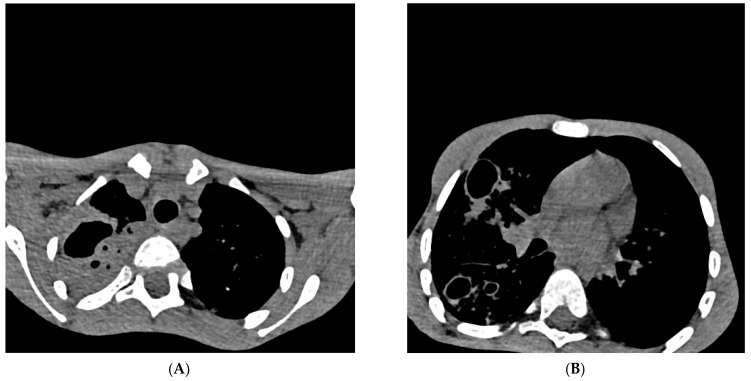
Cross-section of a multislice chest scanner (MSCT). Chest MSCT native and with iv contrast with MPR (28th day of hospitalization), Section 54/209 ((**A**) in Figure 2) and Section 100/209 ((**B**) in Figure 2): on the right, there is a consolidation of the entire upper lobe with signs of excavation (lobar pneumonia) which aroused suspicion, differentially diagnostic of pulmonary tuberculosis. Cystic and traction bronchiectasis are observed in the upper lobe. On both sides of the lung parenchyma, multiple centrilobular nodules (“tree in the bud”) with cystic changes of the thick walls, excavated (obs. Histiocytosis), are more pronounced on the right. Smaller zones of a grouping of spotted consolidation zones are present on both sides in the lower lobes. The trachea and both principal bronchi are passable. Lymphadenopathy of 6 mm paratracheal and established bronchiectasis seen on MSCT of the chest are not relevant exclusively for pulmonary tuberculosis and can already be considered differentially diagnostically in many other respiratory and lung diseases [14]. Pleural spaces are free. The hematological examination of blood smears and myelogram confirmed the infection and rejected the hypothesis of malignant blood disease and solid tumor on the 29th day of hospitalization. Cardiac and ultrasound examination of the heart raised suspicion of infectious endocarditis (mild dysplasia of mitral vela prolapse and a small thickening of 1 × 2 mm in the upper third). Ultrasound examination of the abdomen revealed normal findings, as well as a gentle digital rectal examination. The boy was treated according to protocols for post-COVID-19, sepsis, and pneumonia with antibiotics from the group of reserve antibiotics, mainly parenteral: meropenem + vancomycin 7 days, ciprofloxacin 9 days with concomitant inhalation of colomycin (total 18 days), piperacillin/tazobactam 4 days, vancomycin 8 days, azithromycin 3 days, then, simultaneously for 3 days fluconazole, amoxicillin, and metronidazole, and then, simultaneously for 6 days trimethoprim/sulfamethoxazole and caspofungin. In addition to antibiotics, an antipyretic (ibuprofen, paracetamol) was applied all the time. The systemic corticosteroid methylprednisolone was administered for 20 days, and one dose of normal human immunoglobulins for intravenous use (IgM 6 mg, IgA 6 mg, IgG 38 mg) was administered to treat “post-covid19 inflammation” [15,16]. The patient was referred to IMC, where four antituberculotics (isoniazid, rifampicin, pyrazinamide, ethambutol—H,R,Z,E) were introduced “ex juvantibus” (trial), after which clinical improvement occurred. During the prolonged/post-COVID-19 period, pulmonary tuberculosis can progress “quietly” and “clinically unconvincingly”, which delays diagnosis and delays initiation of adequate treatment. Parents of a sick child may ask you “Why did you introduce four antituberculosis drugs for my child, and you have no evidence that it is tuberculosis?” In the same way, parents can come after a month of treatment with antituberculosis drugs and ask “Why are you excluding antituberculosis drugs now?” “So it’s some other disease after all?”. A new diagnostic test for pulmonary tuberculosis needs to be considered, especially since the eradication of tuberculosis in the world has failed for many years. It is necessary to conduct research on better and more reliable diagnostic tests (microbiological, immunological, or any new type of analysis).

## Figures and Tables

**Table 1 diagnostics-13-02826-t001:** Relevant hematological, biochemical, immunological, and microbiological results in an 11-year-old boy on admission and during 36 days of hospitalization.

Results	Admission	In the Meantime	Released 35th Day	IMC 36th Day
SE 1 h	41	125, 110	70	
CRP (mg/L)	75	72, 68, 84, 36, 60, 59, 49, 30, 66, 91, 62	70	
PCT (ng/mL)	2.6	1.7, 0.9, 0.3, 0.4, 0.1, 0.1, 0.2, 0.3, 0.4, 0.4	0.44	
Le (×10/9 L)	9.8	11.1, 8.7, 6.7, 11.8, 16.4, 16.8, 18.2, 11.6	12.8	
*n* (%)	72	71, 69, 57, 81, 88, 74, 85, 75	68	
ly (%)	17	18, 16, 31, 13, 8, 14, 9, 15	20	
Tr (×10/9 L)	491	545, 454, 474, 503, 394, 414, 432, 423	433	
hemoglobin (g/L)	108	106, 96, 104, 106, 104, 103, 107, 98	97	
LDH (IU/L)	774	868	815	
Feritin (g/L)	200	194, 259, 118	113	
IL6 (pg/mL)	33	31, 203, 15, 18	28	
INR	1.04	1.2, 1.3, 1.3, 1.1, 1.2	1.4	
d-dimer (ng/mL)	15.8	2.7, 2.3, 1.4, 2.0, 1.6, 1.4, 1.3, 1.9	1.9	
Fibrinogen (g/L)	2.8	4.7, 5.2, 4.3, 4.6, 2.1	1.4	
Troponin (mcg/L)	<0.01	<0.01		
IgA (g/L)	5.1		4.6	
IgG (g/L)	18.5		14.8	
IgM (g/L)	2.0		1.7	
IgG-SARSCoV2 (BAU/mL)	40.6			
PCR-respiratory panel	∅	∅, Adenovirus + RSV		
IgM-mycopl.pn.		∅		
Tracheal aspirate	∅	∅, ∅	∅	
Blood culture I	Coag.nSt (from the skin)	∅, ∅, ∅, ∅, ∅, Str.virid, ∅, ∅, ∅	∅	
Blood culture II	∅	∅, ∅, ∅, ∅, ∅, ∅, ∅, ∅, ∅	∅	
Feces on Clostridium difficile	∅		∅	
Swab of the left ear		*S. aureus*		
Mannan test		∅		
Galactomannan test		∅		
QuantiFERON-TB Gold			∅	
Tracheal aspirate and gastric washings-BK (Ziehl–Neelsen)		no found I, II, III		IV no found
Tracheal aspirate and gastric washings-Lowenstein		culture I positive culture II, III negative		IV positive
Tuberculin skin test (TST)				∅
Fiberaspirate (Ziehl–Neelsen, (BD BACTEC MGIT 960)				∅
Fiberaspirate-pathohistological f.				∅

Abbreviations: SE, sedimentation; CRP, C-reactive protein; PCT, procalcitonin; Le, leukocytes; n, neutrophils; Tr, platelets; LDH, lactic dehydrogenase; IL6, interleukin 6; Ig, immunoglobulin; IMC, Institute for Mother and Child New Belgrade; RSV, Respiratory syncytial virus; Coag.nSt, coagulase-negative staphylococci; *S. aureus*, *Staphylococcus aureus*; Fiberaspirate (Ziehl–Neelsen), fiberaspirate after staining by Ziehl–Neelsen; f., finding, ly, lymphocytes.

## Data Availability

Data associated with this study has been deposited at the electronic archive of the University Clinical Centre Pediatric Clinic in Kragujevac, Serbia.

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
