# Peer review of "The Difficulty Detecting Tuberculosis in a Child with Post-COVID-19 and Cerebral Palsy"

_diagnostics, 2023, doi:10.3390/diagnostics13172826_

Round 1

Reviewer 1 Report

Dear Editor,

Currently, numerous studies have shown that tuberculosis after recovering from COVID-19 is becoming more common. The authors have created an interesting Images article, but I think that some clarifications should be made:

Did the patient have lymphopenia on admission?

What was the gamma globulin value?

Tuberculin skin test, tracheal aspirate, gastric washings-BK and fiberaspirate were performed after administration of treatment with Ciprofloxacin - an antituberculous drug?

What did the initial lung CT show, when the patient was admitted?

Thank you!

Author Response

Response to Reviewer 1

1/ On admission, lymphopenia did not exist because the absolute number of lymphocytes was 1998 or 17%, which is now added to the table. /Thomas F. Michniacki, Kelly J. Walkovich. Leukopenia. In: Kliegman RM, St.Geme III JW, Blum NJ, Shah SS, Tasker RC, Wilson KM, Behrman RE. Nelson Textbook of Pediatrics. Elsevier 2021:4770-4790./

2/ The second question is not clear because I did not write "gamma globulin value" anywhere. If the reviewer means reference values for the basic classes of immunoglobulins, then it is as follows: for IgA 0.7-4.0 g/l, for IgG 7-16 g/l, for IgM 0.4-2.3 g/l.

3/ Treatment with ciprofloxacin was carried out 8-16 days after the beginning of hospitalization, this means it was 14 days before taking a sample of tracheal aspirate and gastric washings for direct microscopy for tuberculosis bacilli (which was performed on the 28th day of hospitalization). Tuberculin skin test and fiberaspirate were performed 20 days after the end of ciprofloxacin therapy (which was on the 36th day of hospitalization).

4/ The initial lung MSCT was performed on the 28th day of hospitalization. MSCT of the chest was not completed before admission to the pediatric clinic.

After the expert review, I will send the manuscript for proofreading in English.

I am willing to accept the reviewer's additional suggestion.

Thank you!

Reviewer 2 Report

Comments to the authors:

1. Extensive editing of English language is required. It was really difficult to understand the content.

2. The document presented in this form is not suitable for review. There is no define structure of an article - introduction, method, case presentation, discussions, conclusions. The article begins with legend of table 1.

3. Major revision on imaging. The legends of figure 1 and 2 are inappropriate  with the images itself.

4. Major errors in the medical information provided by the authors.

1. Extensive editing of English language is required. It was really difficult to understand the content.

Author Response

Response to Reviewer 2

1/ After the expert review, I will send the manuscript for proofreading in English.

2/ In the instructions to the authors, it is written that "interesting pictures" should be written as an unstructured manuscript. The applied paper contained an introduction that the Editor asked to be deleted before sending it for review. If the Editor agrees, I will add one paragraph as an introduction to this manuscript, which would look like this:

  1. Introduction

            The prolonged or post-covid19 is accompanied by immunosuppression and catabolic syndrome which is a risk factor for viral reactivation and/or development of secondary infection, among others pulmonary tuberculosis. However, despite all the achievements of modern medicine, it is still difficult to identify the pathogen of tuberculosis, and there is no immunological and radiological confirmation of pulmonary tuberculosis in immunocompromised patients with the prolonged/post-covid19 syndrome. There is a special and high risk of prolonged/post-covid19 syndrome in bedridden patients because they are prone to hypostatic pneumonia, regardless of their immunocompetence. These are often children with spastic quadriplegic cerebral palsy (SQCP) with a level of five loss of rough motor functionality. According to the World Health Organization, about 15,405 people have died of covid19 in Serbia to date, and globally more than 6 million people, provided that about 10% of children aged 2-11 and 13% of children aged 12-16 suffered from the prolonged/post-covid19 syndrome.

3/ Relevant professional information is written right below the picture. In the continuation of the text, an explanation of the clinical picture, laboratory, microbiological diagnostics, and therapy related to the relevant period when the X-ray or MSCT lung was performed is written. In the instructions to the authors for "interesting pictures" as well as in the published "interesting pictures", the content under the pictures is designed exactly like this. If the Editor makes a suggestion to separate the legend from the rest of the accompanying text, I will do so.

4/ It is not clear to me what the reviewer means when he says "Major errors in the medical information provided by the authors". Please write me what "errors" you mean and in connection with what?

I am willing to accept the reviewer's additional suggestion.

Thank you!

Round 2

Reviewer 1 Report

Thanks to the authors for the clarifications. I believe that the article can be published in this form.